# The Effectiveness and Sero-Immunity of Hepatitis B Vaccination in People Who Use Drugs: A Systematic Review and Meta-Analysis

**DOI:** 10.3390/vaccines12091026

**Published:** 2024-09-08

**Authors:** Valeria Reynolds-Cortez, Juan-José Criado-Álvarez, Vicente Martinez-Vizcaino, Carlos Pascual-Morena, Ana Salinas-Vilca, Irene Sequí-Domínguez

**Affiliations:** 1Health and Social Research Centre, Universidad de Castilla-La Mancha, 16002 Cuenca, Spain; valeria.reynolds@alu.uclm.es (V.R.-C.); carlos.pascual@uclm.es (C.P.-M.); irene.sequidominguez@uclm.es (I.S.-D.); 2Preventive Medicine, Hospital Virgen de la Luz, 16002 Cuenca, Spain; 3Institute of Health Sciences, 45600 Talavera de la Reina, Spain; jjcriado@jccm.es; 4Faculty of Health Science, Universidad de Castilla-La Mancha, 45600 Talavera de la Reina, Spain; 5Facultad de Ciencias de la Salud, Universidad Autónoma de Chile, Talca 3460000, Chile; 6Facultad de Enfermería, Universidad de Castilla-La Mancha, 02071 Albacete, Spain; 7Unidad de Calidad, Hospital Virgen de la Luz, 16002 Cuenca, Spain; asalinasv@sescam.jccm.es

**Keywords:** immunization, hepatitis b virus, immunity, drug users, systematic review, meta-analysis

## Abstract

Hepatitis B virus (HBV) vaccination has been available for over four decades. However, a synthesis of the evidence regarding the effectiveness of this strategy on preventing hepatitis B infection in people who use drugs (PWUD) is lacking. A systematic search of the MEDLINE (via PubMed), SCOPUS, Web of Science, and Cochrane Library databases was conducted up to June 2024. Eight studies reported on the effectiveness of HBV vaccination, defined as a positive result for HBsAg or anti-Hbc in vaccinated versus non-vaccinated PWUD, with a pooled effect size of 52% (95% CI: 28.2–67.9) for HBsAg and 31.89% (95% CI: 14.8–45.5) for anti-Hbc. For sero-immunity, defined as the proportion of vaccinated PWUD with levels of anti-HBs ≥ 10 mIU/mL, we found that 66.2% (95% CI: 0.61–0.71; I2 = 94%) had protective levels of anti-HBs. The results of this meta-analysis indicate that the incidence of HBV infection is lower in vaccinated PWUD than in those who did not receive the vaccine. However, the effectiveness is lower than that observed in the general population. This highlights the need for a thorough review of the factors influencing the prevention of HBV infection in PWUD.

## 1. Introduction

The public health impact of hepatitis B (HBV) infection remains a major global concern. This infection has multiple modes of transmission, including sexual contact, vertical transmission, and percutaneous exposure to blood. The diversity of potential exposures poses a significant challenge for the development of effective prevention programs [1].

HBV infection can manifest in a variety of ways, including as an acute hepatitis, liver failure, chronic hepatitis, cirrhosis, portal hypertension, or hepatocellular carcinoma. Worldwide, the prevalence of chronic HBV infection is estimated to be 316 million in 2019, with 555,000 deaths attributed to HBV-related diseases. Furthermore, HBV remains the leading cause of death from liver cancer [1,2].

These current results are despite the availability of interventions aimed to prevent, detect, and treat this infection. The implementation of preventive interventions, such as vaccination, has resulted in an 83% reduction in the global incidence by 2020. However, the burden is concentrated in vulnerable populations, such as drug users, especially in regions with low HBV prevalence (<2%) such as the European Union (EU), where 55% of acute HBV infections in 2020 were attributed to injecting drug users (IDU) [3,4,5,6]. 

IDU represent a well-documented population at risk of HBV infection, mainly due to the practice of sharing injection equipment. The transmission routes for non-injecting drug users are less well defined. It has been postulated that high-risk sexual practices or percutaneous or mucosal exposure to blood-contaminated non-injecting drug equipment, such as crack pipes, may be a potential source of infection. An additional rationale for studying this population is their increased likelihood of transitioning to injecting drug use [7,8].

The efficacy, cost-effectiveness, and safety of HBV vaccination in the general population have been evaluated since its appearance in the early 1980s. These evaluations have focused on two key outcomes: sero-immunity, which is assessed by measuring the seroconversion of hepatitis B surface antigen-specific antibodies (anti-HBs), and the reduction in HBV infection [9,10,11,12]. Despite the proven effectiveness of this intervention, programs targeting people who use drugs (PWUD) remain limited and vaccination strategies vary among countries [5,13]. 

Nevertheless, in the absence of post-vaccination monitoring, the immediate response and long-term protection of the individual to the vaccine remain unknown. The United States Advisory Committee on Immunization Practices (ACIP) currently recommends serological testing in cases where further risk factors are present, such as the presence of certain comorbidities or being a sexual partner of an HBsAg-positive individual. However, in those cases of individuals exposed to an HBsAg-positive source, the appropriate course of action would be determined based on the results of a post-vaccination test. The European Consensus Group on Hepatitis B Immunity states that because of the difficulties of managing this group, post-vaccination testing should only be considered if the user has reduced immunocompetency (e.g., a positive test for HIV), as well as if currently there is no evidence to support booster vaccinations. This last assertion highlights the necessity for further investigation into the effectiveness of vaccination in this group [14,15].

Previous systematic reviews have examined anti-HB sero-immunity in response to HBV vaccination in both IDU and non-IDU populations [16,17,18,19]. However, despite the fact that more than two decades have passed since the introduction of HBV vaccination in this population, a synthesis of the evidence on its effectiveness, assessed as the incidence of HBV infection in vaccinated PWUD, has yet to be undertaken. Therefore, the aim of this systematic review and meta-analysis was to synthesize the evidence on the effectiveness of HBV vaccination in PWUD, both in terms of anti-HB levels and the incidence of HBV infection.

## 2. Materials and Methods

### 2.1. Registration 

This systematic review and meta-analysis was conducted in accordance with the Preferred Reporting Items for Systematic Review and Meta-analysis guidelines (PRISMA) and the Cochrane Handbook for Systematic Reviews of Interventions [20,21]. In addition, the study was previously registered in PROSPERO (CRD42024546749).

### 2.2. Search Strategy

We systematically searched the MEDLINE (via PubMed), SCOPUS, Web of Science, and Cochrane Library databases from their inception to June 2024. In order to be included in this review, the studies retrieved had to meet the following inclusion criteria according to the PI(E)COS strategy: (1) participants (PWUD, regardless of the route of administration, without age limits); (2) intervention or exposure (more than one dose of HBV vaccine); (3) comparison (for incidence of HBV infection, PWUD non vaccinated); (4) outcome: (1) incidence of HBV infection (presence of antibody to hepatitis B core antigen (anti-Hbc) and/or HBsAg measured by standardized microbiological tests) and (2) sero-immunity generated by HBV vaccination (proportion of vaccinated participants with anti-HBs ≥ 10 mIU/mL).

The detailed search strategy for each database is presented in Appendix A. Additionally, we conducted a comprehensive search of the reference lists of the included studies, as well as the gray literature through OpenGrey and Google Scholar. The identified studies were then transferred to the web-based version of Rayyan Systematic Review Tool (new.rayyan.ai, access on 07 January 2024), for further processing. The search was carried out independently by two authors (VR-C and IS-D).

### 2.3. Selection Criteria

The initial step involved the use of Rayyan’s deduplication system to identify and remove all duplicate articles with the same Digital Object Identifier (DOI). This was followed by an individual evaluation of the remaining possible duplicates. Subsequently, the articles were evaluated based on their titles and abstracts to determine their eligibility. 

The inclusion criteria were as follows: (1) observational studies (cross-sectional, case–control, and prospective/retrospective cohort) and intervention studies; (2) exposure or intervention: studies registering one or more doses of HBV vaccine documented or reported; and (3) outcome measure: the prevalence of HBV infection, measured as the presence of anti-HBC and/or HBsAg, and immunity and sero-immunity measured as anti-HBs levels ≥ 10 mIU/mL. Exclusion criteria were (1) studies that included vaccination as a post-exposure prophylaxis, (2) if vaccination was co-administered with hepatitis B immunoglobulin (3), and those that did not recall vaccination history.

Study selection was carried out individually by two authors (VR-C and IS-D), and disagreements were solved by consensus or by a third author (VM-V).

### 2.4. Data Extraction

All 79 articles that met the criteria for full-text review were downloaded into Zotero V 6.0.36 reference management software. An ad hoc table was created with the following data extracted from the included studies: (1) country, (2) study design, (3) population characteristics (age, sex, vaccinated population, comorbidities, substances used, and administration route), (4) vaccination (schedule, dose, brand, and information source), (5) length of follow-up, (6) HBV infection or sero-immunity, and (7) diagnostic test performed in the study. Data extraction was carried out independently by two authors (VR-C and IS-D), and disagreements were solved by consensus or by a third author (VM-V).

### 2.5. Risk of Bias Assessment

For observational studies, we used the Risk of Bias In Non-randomized Studies—of Exposures (ROBINS-E) tool. This tool consists of 7 domains (confounding, measurement of exposure, selection of participants, post-exposure interventions, missing data, measurement of outcomes, and selective reporting of results), which are assessed as “Low”, “Some concerns”, “High”, and “Very high” according to their individual risk of bias. Overall bias was considered “low” if the study was rated as “low risk” in all domains; “some concerns” if there was at least 1 domain with the rating “some concerns”; “high risk” if there was at least 1 domain with the rating “high risk” or several domains with a rating of “some concern”; and “very high risk” if there was at least 1 domain with the rating “very high risk” or several domains at a “high risk”.

For non-randomized intervention studies, we used the Risk of Bias In Non-randomized Studies—of Exposures (ROBINS-I) tool. This tool consists of 7 domains (confounding, measurement of exposure, selection of participants, post-exposure interventions, missing data, measurement of outcomes, and selective reporting of results), which are assessed as “Low”, “Moderate”, “Serious”, “Critical”, and “No information” according to their individual risk of bias. Overall bias was considered “low” if the study was rated as “low risk” for all domains; “moderate” if all domains were rated “low risk” or “moderate”; “serious risk” if there was at least 1 domain with the rating “serious risk”; and “critical risk” if there was at least 1 domain with the rating “critical risk”.

Risk of bias was assessed independently by two reviewers (VR-C and IS-D), and disagreements were solved by consensus or by a third reviewer (VM-V).

### 2.6. Data Synthesis and Statistical Analysis

The included studies were qualitatively summarised in an ad hoc table describing the article characteristics and outcome (i.e., sero-immunity or incidence of HBV infection).

To estimate a random-effects model for the sero-immunity, defined as the proportion of participants with anti-HBs ≥ 10 mIU/mL and with a 95% confidence interval (95% CI), an inverse variance method was used [22]. The effectiveness of the vaccine against HBV infection (1-prevalence ratio) and the 95% CI were calculated using the Mantel–Haenszel method, since it may be preferable to the inverse variance method when data are sparse [23,24].

Heterogeneity was assessed using the I2 statistic, classified as not important if <30%, moderate if 30–50%, substantial if 50–75%, considerable if >75%, and considered statistically significant if *p* < 0.05 [25]. Publication bias was assessed visually using funnel plots and the Egger test, and considered statistically significant if *p* < 0.1 [26,27]. Finally, sensitivity analyses were performed using influence analysis.

A subgroup analysis was conducted to compare the results according to the diagnosis test chosen, the type of microbiological test performed, time since vaccination, vaccination schedule, and a meta-regression of age. All statistical analyses were performed using R-studio version 4.4.0 and Comprehensive Meta-Analysis version 4.0.

## 3. Results

### 3.1. Study Selection and Characteristics

The search identified 16,053 articles, of which 6850 were duplicates, resulting in 9203 unique records. After reviewing the titles and abstracts for compliance with the inclusion and exclusion criteria, 79 full-text articles were assessed for eligibility (Figure 1). Of these, 33 were included in the systematic review and meta-analysis, while 44 were excluded after checking the inclusion/exclusion criteria. One of the articles was included both in the effectiveness and sero-immunity evaluation.

Finally, the list of the excluded articles can be found in Appendix A, which provides detailed reasons for their exclusion.

The reports were published between 1991 and 2020 and included data from 16 countries from Europe, Asia, America, and Oceania. According to the study design, the included reports consisted of 16 clinical trials, and 11 cross-sectional, 4 prospective, 1 cohort, and 1 case–control study. A total of 28 studies were conducted with outpatients, while 5 studies were conducted with inpatients. Notably, only one study reported including participants younger than 18 years of age. The outcomes were evaluated using dried blood spots, saliva, and serological tests.

### 3.2. Risk of Bias Assessment

According to the ROBINS-E and ROBINS-I tools, 11 out of 33 studies (33.3%) were scored as having a “low risk” of bias, and 22 were scored as with “some concerns” or being “moderate risk” studies (66.7%). The detailed risk of bias of each study included is shown in the electronic Appendix A. The main factor contributing to a higher risk of bias in most of the studies was the absence of documentation regarding vaccination.

### 3.3. Effectiveness of HBV Vaccination in People Who Use Drugs

A total of eight studies (Table 1) involving 9771 participants reported on the effectiveness of HBV vaccination in PWUD, one of which reported on two different groups. The pooled estimates of the prevalence of HBV infection, as indicated by the presence of positive serological markers in vaccinated vs. non-vaccinated PWUD, were 15.7% and 25%, respectively.

The meta-analysis included all articles from the systematic review and found a pooled effect size (ES) of vaccine effectiveness of 52% (95% CI: 28.2–67.9) when the outcome was a positive HBsAg result and 31.89% (95% CI: 14.8–45.5) when the outcome was a positive anti-Hbc result (Figure 2).

No significant publication bias was detected based on the Egger’s test (*p*  =  0.93) and the visual inspection of the funnel plot (Appendix A). Furthermore, the trim and fill procedure showed that the removal of the study by Lamden et al. resulted in a reduction in the ES and heterogeneity. 

Subgroup analysis comparing the estimates of saliva and dried blood spots with those of the serological tests (gold standard) did not show statistically significant differences (Appendix A). Similarly, a meta-regression analysis did not find that age influences the ES (Appendix A).

### 3.4. Sero-Immunity Response to HBV Vaccination in People Who Use Drugs

Twenty-six studies (Table 2) involving 5813 participants reported on the proportion of vaccinated PWUD with anti-Hbs protective levels. The pooled proportion of sero-immunity (anti-Hbs ≥ 10 mIU/mL) was 66.2% (95% CI: 0.61–0.71; I2 = 94%; Appendix A).

A significant publication bias was detected based on the Egger’s test (*p* < 0.000) and the visual inspection of the funnel plot (Appendix A). The trim and fill procedure performed indicated that the ES estimate remained unchanged, and thus no correction was performed.

Subgroup analysis comparing the time (months) since vaccination (Figure 3) showed a sero-immunity of 68.7% (95% CI: 62.3–75; I2 = 96.1%) during the first 12 months, and 52.6% (95% CI: 37.6–67.7; I2 = 24.9%) after 12 months [36,37,38,41,42,43,46,47,48,50,51,52,53,54,55,56,57,58,59,60]. Similarly, the subgroup analysis according to vaccination schedule (Appendix A) demonstrated a slightly higher sero-immunity (0.69; 95% CI: 0.62–0.75; I2 = 95.4%) with a standard schedule than with an accelerated schedule (0.62; 95% CI: 0.40–0.73; I2 = 19.1%) [36,37,38,40,41,42,43,45,47,48,50,51,52,54,55,56,57,58,60]. The difference in both subgroup analyses was not statistically significant.

## 4. Discussion

Despite nearly four decades of HBV vaccination, this is, to our knowledge, the first systematic review and meta-analysis to synthesize the evidence on its effectiveness in PWUD. The effectiveness of vaccination in the population of PWUD was 52% (95% CI: 28.2–67.9) for active infection markers and 31.89% (95% CI: 14.8–45.5) when considering the possibility of a resolved infection at the time of testing [61].

According to the Centers for Disease Control and Prevention (CDC), the effectiveness of an adequate schedule for the HBV vaccine in the general population is estimated to range from 75% to 100% [62]. When considering our results, the discrepancies may be attributed to various factors, such as the lack of documentation regarding vaccination, which opens the possibility of an inadequate or incomplete vaccination schedule, vaccine escape mutants, or unknown comorbidities of the participants including liver disease, immunosuppression, and renal disease [63,64,65,66,67,68]. 

Regarding the results of a positive anti-Hbc test, their interpretation is controversial. They could be derived from an exposure to HBV that did not result in infection, occult HBV infection, chronic infection, or a resolved infection [61,69,70]. In order to correctly interpret this result and fully assess the effectiveness of HBV vaccination, we recommend that future studies should employ all serological markers available and measure HBV DNA [71].

Regarding serological immunity, 66.2% of the participants exhibited anti-HBs levels of 10 mIU/mL or greater. These levels are similar to those reported in previous meta-analyses, with the exception of the study by Van Den Ende et al., which reported 65% seroprotection levels with a standard schedule and 58% with an accelerated schedule [16,17,18,19]. Furthermore, antibody levels in our study exhibited a decline over time, in accordance with the current evidence [72]. 

The clinical interpretation of anti-HBs levels over an extended period is currently under debate. The available data indicate that immune memory persists for over 30 years following immunization, and that any subsequent exposure would result in an anamnestic response [12,72]. As previously described, serologic testing after vaccination is not currently indicated in clinical practice, but our results suggest that the incidence of HBV infection in vaccinated PWUD is not negligible [14,15]. Therefore, our data suggest that serologic follow-up should be considered at least on a case-by-case basis. 

Both effectiveness and sero-immunity are commonly used as metrics for evaluating the impact of vaccination in disease prevention, one at the public health level and the other in clinical settings. Our analyses suggest that vaccination helps prevent clinical hepatitis B infection, as expected; however, the level of protection observed appears to be lower than anticipated. Therefore, to ensure the effectiveness of vaccination in preventing HBV infection, it is imperative to thoroughly investigate all cases of HBV infection to ascertain vaccination history in order to detect breakthrough infections in vaccinated PWUD. 

### Limitations

This meta-analysis has several limitations to be acknowledged, some of which are inherent to all meta-analyses (e.g., selection bias and the limited availability of complete information from study reports) and others that are specific to this study. First, the effectiveness of HBV vaccination in PWUD has not yet been sufficiently evaluated, resulting in a limited body of evidence being available. Second, most studies included did not request for documentation to support vaccination, which introduces the possibility of recollection bias, and the possibility of incomplete or inadequate vaccination in most of the participants, which would underestimate the effectiveness of this measure. Third, some studies do not report the serological test used for the diagnosis of HBV infection or the cutoff points for considering sero-immunity; therefore, these studies had to be excluded, which limits the available evidence. Fourth, there are limited data on younger age groups, which represents a significant gap given that these individuals are vaccinated from infancy and information on this cohort could be highly pertinent for evaluating the necessity of follow-up in risk groups. Fifth, most studies evaluating the effectiveness of the HBV vaccination did not report the time elapsed from vaccination to evaluation, and when this information was available, the period was usually shorter than one year. This is crucial information for determining the need and timing of booster doses. Finally, discrepancies in the characteristics of the study samples, analytical tests employed, vaccination schedules and doses, geographic locations, and the quality of the included data may have contributed to increased heterogeneity between studies, particularly in the section of sero-immunity.

## 5. Conclusions

The results of this meta-analysis suggest that, as expected, the incidence of HBV infection in PWUD is lower in those who have been vaccinated against hepatitis B. However, cases of breakthrough infection do occur, and further investigation is needed. In addition to vaccine failure, potential explanations for breakthrough infections include the emergence of vaccine-resistant strains or the presence of comorbidities that may compromise the immune response. Similarly, as no statistically significant difference was identified between accelerated and standard schedules, the value of accelerated schedules, which entail an increase in visits and costs, must be re-evaluated to ascertain their true impact.

Furthermore, it is imperative to continue to focus efforts on the prevention of HBV infection in PWUD through the screening, surveillance, and vaccination of susceptible individuals. Moreover, this review highlights the necessity for a revision of the existing post-vaccination testing protocols in PWUD to detect non-responders and breakthrough infections, and to determine the clinical relevance of post-vaccination antibody levels.

## Figures and Tables

**Figure 1 vaccines-12-01026-f001:**
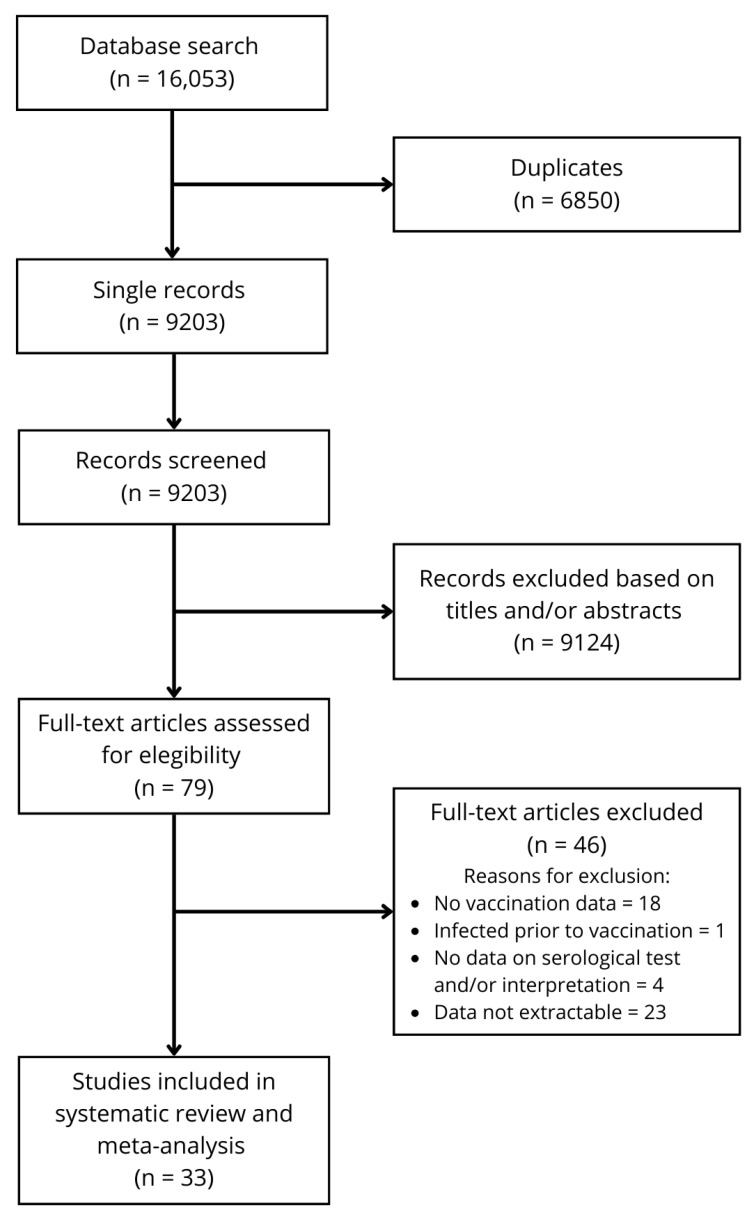
Flow diagram of study selection.

**Figure 2 vaccines-12-01026-f002:**
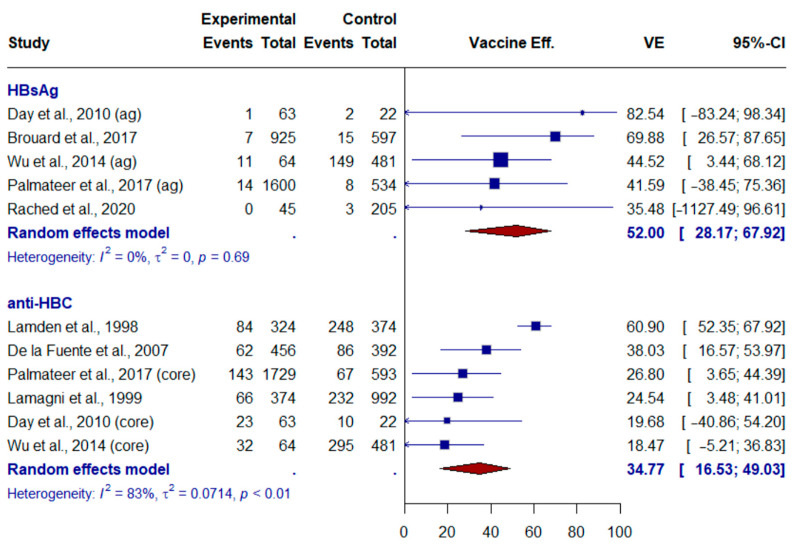
Hepatitis B virus infections in vaccinated and non-vaccinated people who use drugs by serological marker.

**Figure 3 vaccines-12-01026-f003:**
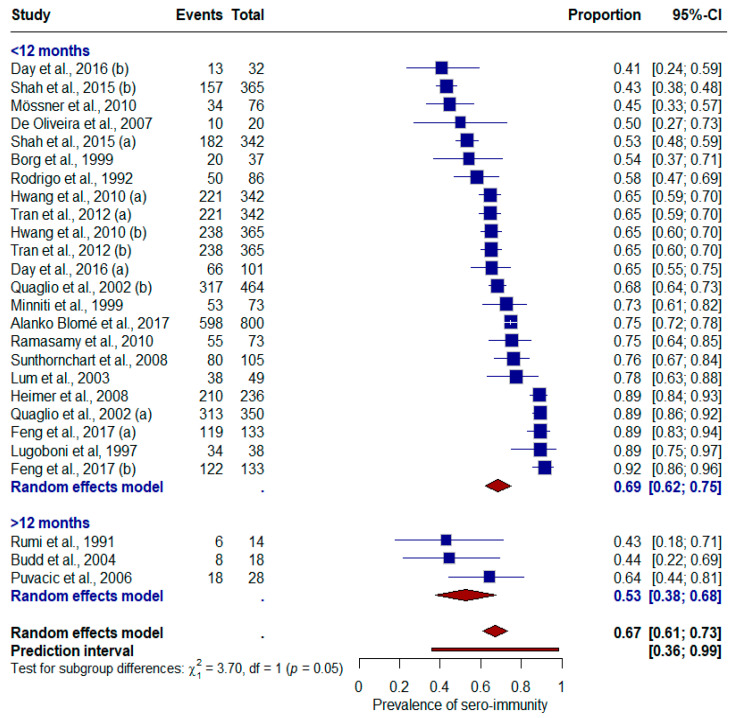
Proportion of sero-immunity (anti-HBs) in people who use drugs by time since vaccination.

**Table 1 vaccines-12-01026-t001:** The characteristics of studies included in the systematic review and the meta-analysis of the effectiveness of hepatitis B virus vaccination in people who use drugs.

Author, Year (Reference)	Country	Study Design	Sample Size (*n*), Male (%)	Age, Years (Mean ± SD)	Population	Nº of Doses, Information Source	HBsAg Positive *n* (%)	Anti-Hbc Positive *n* (%)	Test
Brouard et al., 2017 [28]	France	Cross-sectional	1718, 69.1%	Not specified	Outpatients from harm reduction programs; ≥18 years old	Unknown, reported	7 (0.8%)	no data	Dried blood spots
Day et al., 2010 [29]	Australia	Cross-sectional	209, 68%	37.5 ± 10.9	Outpatients from drug use treatment centers; ≥18 years old	Unknown, reported	1 (1.6%)	23 (38%)	Serological test
De la Fuente et al., 2007 [30]	Spain	Cross-sectional	949, 72.7%	25.7	Street-recruited heroin users; 18 to 30 years old	>1 dose, reported	no data	62 (13.6%)	Dried blood spots
Lamagni et al., 1999 [31]	England	Cross-sectional	1366, 76%	31.5 ± 11.5	Outpatients from drug treatment agencies; no age limit	>1 dose, reported	no data	66 (17.6%)	Saliva
Lamden et al., 1998 [32]	England	Cross-sectional	773, 64%	28.1 ± 6.1	Outpatients from infectious disease unit; no age limit	1–4 doses, documented	no data	84 (25.9%)	Serological test
Palmateer et al., 2017 (a) [33]	Scotland	Cross-sectional	>10,000, not specified	Not specified	Outpatients from drug treatment services and street-recruited drug users; no age limit	>1 dose, reported	no data	143 (8.3%)	Dried blood spots
Palmateer et al., 2017 (b) [33]						>1 dose, reported	14 (0.9%)	no data	
Rached et al., 2020 [34]	Lebanon	Cross-sectional	250, 98%	31.9 ± 8.7	Drug users recruited through nongovernmental organizations; no age limit	Unknown, reported	0 (0%)	no data	Serological test
Wu et al., 2014 [35]	China	Cross-sectional	545, 93.8%	38.5 ± 7.8	Inpatients from compulsory detoxification center; no age limit	>2 doses, reported	11 (16.7%)	32 (50%)	Serological test

**Table 2 vaccines-12-01026-t002:** The characteristics of studies included in the systematic review and the meta-analysis of the sero-immunity response to hepatitis B virus vaccination in people who use drugs.

Author, Year	Country	Study Design	Sample Size (*n*), Male (%)	Age, Years (Mean ± SD)	Population (Drug Users Recruited From…); Age Limit	Vaccine Schedule	Nº of Doses, Information Source	Anti Hbs ≥ 10 mIU/mL *n* (%)
Alanko Blomé et al., 2017 [36]	Sweden	Prospective	800, 71.9%	33.7 ± 13.9	Infectious disease clinic; ≥20 years old	Standard ^1^	3 to 6 doses (¥), documented	598 (74.8%)
Borg et al., 1999 [37]	USA	Clinical trial	43, 62.8%	28.1 ± 1	Methadone treatment; no age limit	Standard ^1^	3 doses, documented	20 (54%)
Budd et al., 2004 [38]	Scotland	Clinical trial	101, 62.2%	31.7 ± 5.4	General practice; no age limit	Accelerated ^a^	3 doses (¥), documented	8 (44.4%)
Collier et al., 2015 [39]	USA	Cross-sectional	519, 89,8%	28 ± 2.4	STAHR study; 18 to 40 years old	no data	Unknown, reported	101 (55.5%)
da Silva et al., 2017 [40]	Brazil	Prospective	600, 84.5%	30 ± 2.6	Chemical dependency unit; ≥18 years old	Accelerated ^b^	3 doses, documented	18 (78.3%)
Day et al., 2010 [29]	Australia	Cross-sectional	209, 68%	37.5 ± 10.9	Drug use treatment centers; ≥18 years old	no data	Unknown, reported	33 (52.4%)
Day et al., 2016 (a) [41]	Australia	Clinical trial	139, 77%	32.9 ± 8.2	HITS-c study and needle and syringe programs; ≥16 years old	Accelerated ^b^	3 doses (¥), documented	66 (62%)
Day et al., 2016 (b) [41]						no data	>1 dose (¥), documented	13 (41%)
De Oliveira et al., 2007 [42]	Brazil	Case- control	60, 100%	40.7 ± 10.1	Alcohol dependency treatment; no age limit	Standard ^1^	3 doses (¥), documented	10 (50.0%)
Feng et al., 2017 (a) [43]	China	Clinical trial	480, 100%	37.0 ± 8.7	Drug rehabilitation center; no age limit	Standard ^1^	3 doses (¥), documented	119 (89.5%)
Feng et al., 2017 (b) [43]				36.4 ± 8.6		Standard ^1^	3 doses (π), documented	122 (91.7%)
Grogan et al., 2005 [44]	Ireland	Cross- sectional	316, 60%	29.8 ± 10.1	Addiction treatment centers; no age limit	no data	> 3 doses, documented	114 (85.1%)
Hagedorn et al., 2010 (a) [45]	USA	Prospective	104, 99%	54.5 ± 9.3	Veterans Medical Center Addictive Disorders Service; no age limit	Standard ^1^	3 doses (¥), documented	41 (52.6%)
Hagedorn et al., 2010 (b) [45]						Standard ^1^	3 doses (¥), documented	13 (50%)
Heimer et al., 2008 [46]	USA	Clinical trial	1964, 72.6%	Not specified	Syringe exchange program; ≥18 years old	Accelerated ^a^/standard ^1^	3 doses, documented	210 (89.0%)
Hwang et al., 2010 (a) [47]	USA	Clinical trial	1260, 76%	Not specified	Street-recruited; ≥18 years old	Standard ^1^	3 doses (¥), documented	221 (65%)
Hwang et al., 2010 (b) [47]						Accelerated ^a^	3 doses (¥), documented	238 (65%)
Lugoboni et al., 1997 [48]	Italy	Clinical trial	50, not specified	24.1 ± 4.8	Drug users center; no age limit	Standard ^1^	3 doses (¥), documented	34 (89%)
Lugoboni et al., 2004 [49]	Italy	Prospective	895, 81.1%	Not specified	Public addiction clinics; no age limit	Standard ^1^	1 to 5 doses, documented	230 (71.9%)
Lum et al., 2003 [50]	USA	Clinical trial	170, 71%	21.3 ± 1.5	UFO study; <30 years old	Standard ^1^	3 doses (¥), documented	38 (77.5%)
Minniti et al., 1999 [51]	Italy	Clinical trial	110, 81.8%	39.5 ± 6.3	Drug use treatment center; no age limit	Accelerated ^a^	3 doses (Ω), documented	53 (73,2%)
Mössner et al., 2010 [52]	Denmark	Cross-sectional	235, 73%	40.3 ± 3.6	Drug use treatment centers; no age limit	Accelerated (other)	4 doses (¥), documented	34 (44.7%)
Puvačić et al., 2006 [53]	Bosnia and Herzegovina	Cohort	28, not specified	Not specified	Non-specified; no age limits	no data	3 doses, reported	18 (64.3%)
Quaglio et al., 2002 (a) [54]	Italy	Clinical trial	1175, 83%	25.7	Public health centers; no age limit	Standard ^1^	3 doses (¥), documented	313 (89%)
Quaglio et al., 2002 (b) [54]						Accelerated ^a^	3 doses (¥), documented	317 (68%)
Ramasamy et al., 2010 [55]	Australia	Clinical trial	143, 71.3%	33.1 ± 8.3	Methadone maintenance program; no age limit	Accelerated ^a^	3 doses (Ω), documented	55 (75.4%)
Rodrigo et al., 1992 [56]	Spain	Clinical trial	86, 79.1%	23.7	Section of drug addiction; no age limit	Accelerated ^a^	3 doses (¥), documented	50 (58%)
Rumi et al., 1991 [57]	Italy	Clinical trial	55, 74.6%	25 ± 4.3	Rehabilitation program; no age limit	Standard ^1^	3 doses (¥), documented	6 (43%)
Shah et al., 2015 (a) [58]	USA	Clinical trial	1260, 77%	Not specified	Street-recruited; ≥18 years old	Standard ^1^	3 doses (¥), documented	182 (53.2%)
Shah et al., 2015 (b) [58]						Accelerated ^a^	3 doses (¥), documented	157 (43.1%)
Sunthornchart et al., 2008 [59]	Thailand	Clinical trial	105, not specified	Not specified	Methadone clinics; ≥20 years old	Standard ^1^	3 doses, documented	80 (76.2%)
Tran et al., 2012 (a) [60]	USA	Clinical trial	1260, not specified	43 ± 9	Street-recruited; ≥18 years old	Standard ^1^	3 doses (¥), documented	221 (64.6%)
Tran et al., 2012 (b) [60]						Accelerated ^a^	3 doses (¥), documented	238 (65.2%)

^1^ Standard schedule: 0, 1, 6 months. ^a^ Accelerated schedule: 0, 1, 2 months. ^b^ Accelerated schedule: 0, 7, 21 days. (¥) Dose of 20 mcg. (Ω) Dose of 10 mcg. (π) Dose of 60 mcg.

## Data Availability

The raw data supporting the conclusions of this article will be made available by the authors on request.

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
