# Peer review of "The Effectiveness and Sero-Immunity of Hepatitis B Vaccination in People Who Use Drugs: A Systematic Review and Meta-Analysis"

_vaccines, 2024, doi:10.3390/vaccines12091026_

Round 1

Reviewer 1 Report

Comments and Suggestions for Authors

Thank you to the authors for their willingness to design and execute a novel, rigorously conceptualized study, write it up and submit it for peer review. As with all studies, there are places where the paper can be further strengthened with additional development. However, I found the paper to be thoughtful, well-reasoned, and of potential interest to the readership of Vaccines. My thoughts and suggestions for further development are provided below:

(1)     Study design is appropriate and well-conceived. The sequence of steps used to select relevant studies is described well. Databases accessed are the most appropriate ones. Sufficient attention is devoted to the gray and hard to access literature. MESH headings, titles, key words were all searched.

(2)     Scientific justification is strong and well-detailed across multiple countries. Literature search included experimental, observational, and outcome studies.

(3)     Investigation for potential bias in published studies represents a relative strength of the paper. Authors may wish to resolve the potential discrepancy between the nonsignificant (effectiveness) and significant (sero immunity) inferences. Additional information will be helpful to the reader.

(4)     Study characteristics and effect sizes are important and clearly presented.

(5)     Mantel-Haenszel (MH) is a fine technique for analysis, but the authors may wish to state why they selected the MH technique specifically. For example, DerSimonian-Laird is often held to be the strongest technique for random effects models while the MH technique is often a “go to” method for fixed effects models. Authors may wish to revisit their wording and rationale (see pg 4).

(6)     With respect to sequencing of the study, all parts flow from one another. Conclusions are logical, supported by the findings, and consistent with the prior literature.

(7)     The summary finding that the incidence of hepatitis B infection is lower in those who have received the vaccine is unremarkable. Formally documenting this finding in hard to (or virtually impossible to) access populations is important, but the rather logical nature of the findings should be acknowledged in light of the existing literature. One of the findings that I found most interesting pertained to the subgroup analyses regarding scheduling of the vaccinations and lack of significance between accelerated and nonaccelerated schedules. Would this warrant additional description in the conclusion?

In conclusion, I found the current manuscript to be well-designed, well-written, interesting, and of potential appeal to the readership of Vaccines.

Author Response

Dear reviewer,

Reviewer 2 Report

Comments and Suggestions for Authors

This is a systematic analysis and meta-analysis that includes 16 countries from four continents. It examines the existing evidence regarding the effectiveness of vaccination and seroconversion after vaccination against HBV in patients who use intravenous drugs.

The authors themselves comment on the problems that arise from this study. They are derived firstly from the fact that some of the included studies did not have documentation to support the vaccination history of the participants, which can introduce a bias in the same.

In some studies, the time between vaccination and evaluation was less than a year, which does not allow for evaluating the long-term protection or the need for booster doses and underestimates the duration of immunity and, therefore, the effectiveness of the vaccine over a longer period.

Consider that the heterogeneity of the studies, particularly in seroimmunity, can lead to underestimated values, but this is a problem inherent to meta-analysis

Finally, hepatitis B is a global problem, and in high-risk populations such as intravenous drug users this is more important due to the higher prevalence of infection and the importance of prevention through vaccination. The results derived from the importance of vaccinating this high-risk group, so that it is part of global health policies, underline the urgent need for coordinated global health policies to combat this issue.

The manuscript is clear, the methodology used for the inclusion of the articles, and the biases that could be inherent were subclassified to have more solid conclusions. 

Author Response

Dear reviewer,

Reviewer 3 Report

Comments and Suggestions for Authors

The topic is simple, but interesting and intriguing from a scientific point of view.

In my opinion, the scientific and statistical preparation is very good, due to the lack of a large knowledge of statistics I cannot unequivocally assess the proper selection of tests and methods.

congratulations on the prepared article, there are no substantive comments, rather editorial:

1. key word - use lower case letters

2. citation numbers should be given after a comma, e.g. [1,2] and not [1-2] - to be corrected at least. in line: 48, 60, 249

3. bibliography numbers will be used at the end of the sentence, not in the middle (e.g. line 61, 75, 155, 228, 256, 265)

4. use the abbreviation ACIP in parentheses

5. anty-Hbc instead of anty-HBC (line 114, 250, figure 3)

6. figure 3 - use lowercase letters

7. figure 4 - use lowercase letters or the abbreviation PWUD

8. line 243 - too many dots

9. line 244 - was the abbreviation CDC previously expanded?

10. line 259 - remove the comma

11. line 296 - period or comma?

Author Response

Dear reviewer,
